# IRIDA Phenotype in *TMPRSS6* Monoallelic-Affected Patients: Toward a Better Understanding of the Pathophysiology

**DOI:** 10.3390/genes13081309

**Published:** 2022-07-23

**Authors:** Vera Hoving, Scott E. Korman, Petros Antonopoulos, Albertine E. Donker, Saskia E. M. Schols, Dorine W. Swinkels

**Affiliations:** 1Department of Hematology, Radboud University Medical Center, Geert Grooteplein Zuid 10, 6525 GE Nijmegen, The Netherlands; vera.hoving@radboudumc.nl; 2Translational Metabolic Laboratory, Department of Laboratory Medicine, Radboud University Medical Center, Geert Grooteplein Zuid 10, 6525 GE Nijmegen, The Netherlands; scott.korman@radboudumc.nl (S.E.K.); petros.antonopoulos@ru.nl (P.A.); dorine.swinkels@radboudumc.nl (D.W.S.); 3Department of Pediatrics, Máxima Medical Center, De Run 4600, 5504 NB Veldhoven, The Netherlands; albertine.donker@radboudumc.nl; 4Sanquin Blood Bank, Sanquin Diagnostics BV, Plesmanlaan 125, 1066 NH Amsterdam, The Netherlands

**Keywords:** IRIDA, anemia, iron, heterozygous *TMPRSS6*, monoallelic *TMPRSS6*, phenotype

## Abstract

Iron-refractory iron deficiency anemia (IRIDA) is an autosomal recessive inherited form of iron deficiency anemia characterized by discrepantly high hepcidin levels relative to body iron status. However, patients with monoallelic exonic *TMPRSS6* variants have also been reported to express the IRIDA phenotype. The pathogenesis of an IRIDA phenotype in these patients is unknown and causes diagnostic uncertainty. Therefore, we retrospectively summarized the data of 16 patients (4 men, 12 women) who expressed the IRIDA phenotype in the presence of only a monoallelic *TMPRSS6* variant. Eight unaffected relatives with identical exonic *TMPRSS6* variants were used as controls. Haplotype analysis was performed to assess the (intra)genetic differences between patients and relatives. The expression and severity of the IRIDA phenotype were highly variable. Compared with their relatives, patients showed lower Hb, MCV, and TSAT/hepcidin ratios and inherited a different wild-type allele. We conclude that IRIDA in monoallelic *TMPRSS6*-affected patients is a phenotypically and genotypically heterogeneous disease that is more common in female patients. We hypothesize that allelic imbalance, polygenetic inheritance, or modulating environmental factors and their complex interplay are possible causes. This explorative study is the first step toward improved insights into the pathophysiology and improved diagnostic accuracy for patients presenting with IRIDA and a monoallelic exonic *TMPRSS6* variant.

## 1. Introduction

Iron-refractory iron deficiency anemia (IRIDA) is a rare inherited form of iron deficiency anemia characterized by inappropriately high hepcidin levels relative to body iron status and degree of anemia [1,2,3,4]. It is considered to follow an autosomal recessive inheritance pattern, but patients have been reported expressing the phenotype in the presence of a monoallelic *TMPRSS6* defect [2,3,5,6,7]. The exact mechanism that explains the development of an IRIDA phenotype in these patients is still unknown. To prevent diagnostic uncertainty in these patients, it is important to gain more insight into the pathophysiology.

Iron homeostasis is tightly regulated to ensure a balance between uptake, transport, storage, and utilization [8]. The main regulator in systemic iron homeostasis is hepcidin, a hepatic peptide hormone. It controls cellular iron release by binding and subsequent occlusion or degradation of the cellular iron exporter ferroportin, which is mainly expressed in enterocytes and macrophages [9,10,11,12]. On a molecular level, the expression of hepcidin is regulated via the bone morphogenetic protein (BMP)/sons of mothers against decapentaplegic (SMAD) signaling pathway. Erythropoiesis downregulates hepcidin expression via the interaction of erythroid factors, such as erythroferrone (ERFE) with BMPs [13,14,15]. In contrast, inflammation increases hepcidin expression, mainly via interleukin-6 (IL-6), which induces the Janus kinase (JAK)/signal transducer and activator of transcription 3 (STAT3) pathway. The main hepcidin inhibitor in this BMP/SMAD pathway is matriptase-2 (MT-2), a serine protease encoded by *TMPRSS6* [16]. The mechanism of MT-2-mediated hepcidin synthesis inhibition is not completely understood, but MT-2 has been reported to cleave membrane hemojuvelin (HJV), thereby attenuating BMP-SMAD signaling and hepcidin transcription [17,18,19,20]. The biological relevance of *TMPRSS6* is highlighted by the appearance of IRIDA, which results from homozygous or compound heterozygous *TMPRSS6* variants.

IRIDA is phenotypically characterized by microcytic anemia with low serum iron parameters (serum iron and transferrin saturation (TSAT)) and a discrepantly high level of hepcidin in relation to body iron levels. These elevated hepcidin levels account for the absent or only partial response to oral iron treatment due to the inhibition of intestinal iron absorption [1,9]. IRIDA is considered an autosomal recessive inherited disease, with most patients described as harboring biallelic exonic pathogenic variants in *TMPRSS6* [2,3,21]. However, some patients have been reported expressing the IRIDA phenotype in the presence of only monoallelic exonic *TMPRSS6* defects, although this phenotype is generally less severe compared with biallelic patients [1,2,3].

To define the clinical phenotype of monoallelic *TMPRSS6*-affected patients, we characterized a series of 16 monoallelic IRIDA patients belonging to 15 different families and 8 of their relatives with identical (possible) pathogenic exonic variants without expressing the phenotype. To gain better insight into the genetic differences between patients and their unaffected relatives, *TMPRSS6* haplotype analysis was performed. By using phenotypic and genotypic data of our case series, we explored the possible pathophysiological mechanisms of phenotype expression in these *TMPRSS6* monoallelic-affected patients.

## 2. Materials and Methods

### 2.1. Patients and Relatives

Patients were included based on the results of the genetic analysis of *TMPRSS6,* which was performed for diagnostic purposes in our tertiary referral center. Clinical and genetic patient data were pseudonymized and kept in the Castor EDC database, an electronic online secured platform. Additional information was extracted from the electronic medical record system only from patients who did not enter an opt-out notification in their medical records. All subjects were inhabitants of the Netherlands and were diagnosed between 2009 and 2019. This study was approved by the Medical Research Ethics Committee (MREC) Oost-Nederland (protocol number 2017-3270, date 1 February 2022).

Patient sample selection was based on the diagnosis of a monoallelic IRIDA genotype, defined as the presence of one (possible) pathogenic *TMPRSS6* variant (defined as a class 3, 4, or 5 exonic variant [22,23]) and the presence of an IRIDA phenotype. As evidence-based criteria for an IRIDA phenotype are not available, we used our own expert opinion-based criteria to support the diagnosis. These criteria included the (i) presence of microcytic anemia (MCV < 80 fl; Hb < 12.0 g/dL in women and Hb < 13.0 g/dL in men) in the absence of treatment with oral or IV iron administration, (ii) TSAT < 10% (and < 15% when on oral or IV iron treatment) in the absence of inflammation (CRP < 10 mg/L) unless the treating physician considered the inflammation an unlikely (only) cause of the patient’s phenotype, and (iii) no or only a partial response to oral iron administration [3]. Iron deficiency anemia (IDA) due to other forms of iron-refractory IDA (e.g., autoimmune gastritis, celiac disease, *Helicobacter pylori*, and other gastrointestinal infections) were excluded, if applicable. TSAT/hepcidin ratio was used as an additional criterion for IRIDA phenotype confirmation (TSAT/hepcidin ≤ 2.5 percentile in relation to age and gender supporting the diagnosis) [3,24,25].

### 2.2. Laboratory Testing

Hemoglobin, red blood cell indices, serum iron parameters, and CRP were assessed by ISO-accredited Dutch hospital laboratories. Hepcidin serum levels were assessed at the Radboudumc by weak cation exchange time-of-flight mass spectrometry (WCX-TOF MS) between 2009 and 2019. Since February 2019, the hepcidin assay has been standardized using secondary reference material (RM) that was value assigned by provisional primary RM [26,27]. Reference ranges for this standardized assay are available on our website [25]. Measurements of serum hepcidin were performed both before and after the standardization of the assay. Standardization slightly altered the hepcidin values since standardized results were a factor 1.054 higher compared with historic results obtained without standardization [25]. The stability of the assay in time was ensured by using serum matrix-based quality controls and Westgard rules to evaluate the results [28].

### 2.3. Genetic Analysis

Genotyping was performed by PCR, DNA Sanger sequencing (until March 2014), and Ion Torrent sequencing (after March 2014) of the coding part and the intron–exon boundaries of *TMPRSS6*. Multiplex ligation-dependent probe amplification (MLPA) was conducted to rule out large deletions or duplications and insertions in the wild-type allele (Appendix A) [3,29]. The pathogenicity of the found genetic variants was assessed by a review of the literature on previously reported cases and functional studies, the association of the variant with the phenotype within a family, and by bioinformatic tools (SIFT, Align GVGD, Polyphen, SpliceSiteFinder-like, MaxEntScan, NNSplice, GeneSplicer, and Human Splicing Find, all as part of the Alamut software) [3,22,23,30]. *TMPRSS6* variants were classified according to the joint English and Dutch practice guidelines as either clearly pathogenic (class 5), likely to be pathogenic (class 4), uncertain pathogenicity (class 3), unlikely to be pathogenic (class 2), or clearly not pathogenic (class 1) [22,23].

### 2.4. Haplotype Analysis

To determine the likelihood of intragenic variants being responsible for the IRIDA phenotype, we conducted a family haplotype analysis on the *TMPRSS6* exonic genotype of five monoallelic-affected IRIDA probands and six of their clinically unaffected relatives. Haplotype analysis was performed using information on the presence of nonpathogenic variants in the *TMPRSS6* coding sequences.

## 3. Results

### 3.1. Subject Characteristics

A total of 24 subjects (15 probands and 9 relatives) with only one affected *TMPRSS6* allele were included in the study. In total, 16 subjects from 15 unrelated families showed the IRIDA phenotype, and 8 relatives with the same (possible) pathogenic *TMPRSS6* variant, as their proband did not have an IRIDA phenotype (Table 1 and Table 2).

The majority of clinically affected subjects were women (*n* = 12, 75%), and most of them (*n* = 8, 67%) were genotypically diagnosed with monoallelic IRIDA around the fourth and fifth decade of life, although they presented with microcytic anemia during adolescence. Only 4 male patients were diagnosed with monoallelic IRIDA, 3 of them during childhood (at ages ranging from 2 to 4 years). All affected subjects presented with microcytic anemia, but the severity of the anemia was highly variable (Hb range 5.5–12.6 g/dL, MCV range 55–81 fl) and was independent of the *TMPRSS6* genotype. Ten patients received iron supplementation (both oral or IV) at the time of genotypic diagnosis; in the other 7 patients, treatment started after diagnosis. The majority of the patients (11 out of 17, 69%) required IV iron administration due to unresponsiveness to oral treatment. Only one patient (ID 5) received a transfusion due to postpartum hemorrhage after a cesarian section.

Compared with their clinically unaffected but *TMPRSS6* genotypically identical family members, affected individuals showed lower TSAT/hepcidin ratios (median 1.6%/nM, range 0.3–2.7%/nM versus median 7.2%/nM, range 0.8–12.9%/nM). One relative (ID 24) had a TSAT/hepcidin ratio below the reference range (TSAT/hepcidin 0.8%/nM), but this is not consistent with an IRIDA phenotype due to the absence of microcytic anemia, a TSAT > 15%, and the presence of inflammation (CRP 45 mg/L).

At the time of genotypic diagnosis, monoallelic-affected patients also showed lower values of Hb (median Hb 11.4 g/dL (IQR 10.6–12.4) versus 12.8 g/dL (IQR 11.6–13.4)), MCV (median MCV 78fl (IQR 69–86) versus 88fl (IQR 82–91)), and TSAT (median TSAT 8.8% (IQR 4.8–12.6) versus 18.6% (IQR 14.9–23.4)) than their unaffected relatives with the same *TMPRSS6* pathogenic variant (Table 1 and Table 2).

In our cohort, 4 patients suffered from comorbid conditions (i.e., infection and inflammatory disorders such as Familial Mediterranean Fever syndrome, *Giardia lamblia* infection, recurrent upper respiratory tract infections, and blood loss (uterine myoma)) that probably contributed to the iron deficiency anemia. However, anemia persisted after recovery from these conditions, and, therefore, IRIDA is considered the underlying cause.

Patients carried different genotypes. Both patients and relatives share a variety of nonpathogenic exonic variants. We could not observe combinations of nonpathogenic variants that were exclusively found in patients or unaffected relatives (Appendix A).

Taken together, the IRIDA phenotype in monoallelic *TMPRSS6*-affected patients shows a very heterogeneous clinical picture, whereby environmental factors such as infections and blood loss contribute a significant part to the phenotype.

### 3.2. Haplotype Analysis

Haplotype analysis was performed in five families, including five probands and seven relatives (four first-degree and three second-degree relatives, Appendix A). Patients differed from their unaffected relatives in wild-type allele. Both patients and relatives share a variety of nonpathogenic exonic variants. We could not observe combinations of nonpathogenic variants that were exclusively found in patients or unaffected relatives (Appendix A). 

## 4. Discussion

In this study, we described a cohort of 16 monoallelic *TMPRSS6* patients with an IRIDA phenotype and eight relatives with an identical exonic *TMPRSS6* variant as their proband but without the phenotype. Monoallelic-affected patients differed in their *TMPRSS6*, and expression and severity of the IRIDA phenotype were highly variable. The majority of monoallelic IRIDA patients consisted of female patients. Compared with clinically unaffected but *TMPRSS6* genotypically identical relatives, affected individuals presented with a markedly lower Hb level, MCV, and TSAT/hepcidin ratio. Haplotype analysis showed a difference in wild-type alleles between patients and their unaffected relatives.

In our own tertiary center, the monoallelic-affected patients form around 1/3 of the total population of patients with *TMPRSS6*-related IRIDA, with the other 2/3 consisting of biallelic-affected patients. Since it is conceivable that only the monoallelic patients with a more severe phenotype come to our attention, the percentage of monoallelic-affected patients may be higher in the general population, and it may also depend on the definition of anemia.

Clinical presentation of monoallelic IRIDA in our patients varied from mild microcytic anemia responding to oral iron supplementation to pronounced microcytic anemia requiring parenteral iron therapy. Age at diagnosis of monoallelic IRIDA was very variable between patients. Remarkably, almost all male patients in our cohort were diagnosed in childhood, which is corroborated by previous reports [2], whereas the majority of female patients were diagnosed in adulthood. We observed a variation in the underlying genotype. Despite this genotypic and phenotypic heterogeneity, all of our monoallelic patients shared their need for iron administration. This corresponds with findings of previous reports, in which therapy for *TMPRSS6* monoallelic patients varied from oral iron supplementation to liposomal iron; only one patient required a red blood cell transfusion [5,6,7,31,32].

Our study showed that monoallelic IRIDA is more common in (premenopausal) women, confirming observations in smaller case series (Appendix A) [3,5,6,7,33]. Despite the small number of included subjects, it is conceivable that females are prone to develop an IRIDA phenotype due to the presence of environmental factors, such as menses, pregnancy, or delivery. Another explanation for the difference between the male and female proportions might be explained by hormonal influences. Both estrogen and testosterone have been shown to suppress hepcidin expression. It is plausible that the suppressive effect of testosterone is stronger than that of estrogen, but studies to confirm this are lacking [34,35].

While probands and their relatives in our study share the same (possible) pathogenic exonic variant, only probands presented with an IRIDA phenotype. The severity of the IRIDA phenotype differed between patients and between patients sharing the same pathogenic variant. This suggests the presence of factors that increase the susceptibility of monoallelic patients to express the IRIDA phenotype and its clinical severity. We hypothesized that the difference in expression of the IRIDA phenotype could be caused by an imbalance of *TMPRSS6* in favor of the affected allele, polygenetic inheritance, or the presence of certain disease-modifying environmental factors.

Allelic imbalance could be the result of a genetic defect in the noncoding sequence (i.e., introns, promotor, and 3′ untranslated region (3′UTR)) of the healthy allele that affects its expression and possibly also function. Indeed, in our small study, we found that patients differed from their relatives in wild-type alleles. Sharing the affected allele and not the ‘healthy’ wild-type allele points toward a possible role of the presumed ‘healthy’ wild-type allele in the expression of IRIDA. However, we did not find a particular nonpathogenic variant influencing the disease.

As deep intronic variants are not detected by analyzing only the coding sequences and intron–exon boundaries due to their deep intronic location, we could not exclude this as an underlying cause. Deep intronic variants create cryptic splice sites, resulting in the production of a cryptic exon by the inclusion of an intronic fragment. This might have an impact by causing frameshift or the introduction of a premature stop codon, leading to nonsense-mediated mRNA decay or protein truncation [36]. Overall, deep intronic variants in OMIM genes are uncommon but not rare, and when inherited *in trans* with pathogenic exonic variants, it is conceivable they may lead to an IRIDA phenotype.

Next to deep intronic variants, variants in other noncoding sequences may also cause allelic imbalance. Different approaches have been developed to assess this imbalance in genes. In a study in which they used an array-based method to assess this differential allelic expression, they found *TMPRSS6* alleles are differentially expressed (in a 60:40 ratio) and broadly mapped the underlying regulatory haplotype to the 3′UTR [37]. It is plausible that differential expression can contribute to disease in monoallelic IRIDA when it favors the affected *TMPRSS6* allele and reduces the expression of the functional allele. Generally, this provides an explanation for the clinical presentation of monoallelic-affected IRIDA patients to have a greater effect on the phenotype in these individuals [38].

Since we observed that monoallelic IRIDA patients show a milder phenotype compared with biallelic patients [3], we conclude that if variants in noncoding sequences in the presumed healthy allele cause the phenotype in monoallelic-affected IRIDA patients, they do allow some expression of the gene.

Allelic imbalance could also be provoked by epigenetic regulation of the wild-type allele. Although this has not been investigated yet in *TMPRSS6*, it has been reported that different epigenetic regulations (e.g., histone modification, DNA methylation chromatin remodeling) play an important role in other iron sensing genes to adapt to conditions and stressors that determine the transcription of hepcidin, such as iron deficiency, inflammation, or hypoxia [39,40,41,42]. Furthermore, epigenetic modifications deposited in the promotor region of *TMPRSS6* itself might lead to changes in its expression.

Another mechanism that may underly the difference in expression of IRIDA phenotype among subjects carrying identical exonic *TMPRSS6* variants involves the expression of quantitative trait loci (eQTL). These eQTLs cause variation in mRNA expression levels but are not disease-causing themselves. eQTLs are categorized into *cis*- or *trans*-acting: *cis*-acting eQTLs are situated within 1Mb of the gene they act on, and *trans*-acting eQTLs are distant from the corresponding gene (> 5Mb) and are often situated on different chromosomes [43]. The influence of *cis*-acting eQTLs on the wild-type allele is probable in monoallelic patients whose healthy relatives carry a different wild-type allele. On the contrary, in patients whose healthy relatives carry identical alleles, *trans*-acting eQTLs are more likely. Indeed, in a performed genome-wide meta-analysis, many loci associated with iron homeostasis biomarkers were predicted to target distant genes (*trans* eQTLs) [44]. For instance, one variant associated with iron levels is situated on *HAMP* (hepcidin antimicrobial peptide) and targets *FRAR2* (free fatty acid receptor 2; rs2005682); another variant associated with iron is situated on *RAB6B* (member RAS oncogene family) and targets *TF* (transferrin; rs4854760). Furthermore, two variants were associated with a higher risk for IDA, namely in *DUOX2* (dual oxidase 2; rs57659670) and one novel variant in *SLC11A2* (solute carrier family 11 member 2) [44]. In light of these findings, it is conceivable that the combination of a single deleterious *TMPRSS6* allele, together with the inheritance of distant-acting eQTLs, might elicit the IRIDA phenotype in patients in contrast to their healthy relatives with the identical *TMPRSS6* allele combination.

Expression and severity of the IRIDA phenotype may also be attributed to alterations in genes other than *TMPRSS6*. The expression of hepcidin is regulated by various proteins and signaling pathways (Figure 1). It is plausible that the loss or gain of function of one or more important upstream modulators of hepcidin expression, the HAMP gene itself, or its downstream effects on ferroportin activity modify the phenotype of patients with monoallelic *TMPRSS6* defects. This is corroborated by the work of Pagani et al., who demonstrated proof of this concept by means of a patient carrying an activin receptor 1A *(ACVR1A)/TMPRSS6* compound heterozygous genotype, presenting with fibrodysplasia ossificans progressiva (FOP) as well as an IRIDA phenotype [45]. In vitro investigation showed that the *ACVR1A* variant (encoding the activin receptor-like kinase-2 (ALK2)) caused the ALK2 receptor to upregulate hepcidin expression, even in the absence of its ligand BMP [45]. In analogy, mutations of other constituents of the upstream hepcidin expression pathways or downstream effect on ferroportin activity could be the reason why a monoallelic IRIDA manifests. However, to comply with the observation of our cohort, in which IRIDA is the predominant disease, such mutations would have to possess certain characteristics: (i) the mutations themselves should not have a significant clinical effect on their own (i.e., not be disease-causing themselves); (ii) they should have an overall stimulating effect within the hepcidin expression pathway or an inhibiting effect on ferroportin activity; and (iii) in combination with a *TMPRSS6* variant, they should resist any attenuation by cellular feedback mechanisms.

A third main potential mechanism of the differential expression of the IRIDA phenotype may be related to exposure to certain environmental factors that might trigger the phenotype expression in monoallelic subjects. This is supported by findings in *TMPRSS6* haploinsufficient mice that are more susceptible to developing iron deficiency under the condition of iron restriction or an increased iron requirement [46]. As mentioned above, several conditions such as inflammation, infection, and elevated iron requirement affect hepcidin levels and body iron status and, thus, are considered modulating factors [18]. Although we excluded these conditions as a main cause of the anemia, they may contribute to the expression and severity of the phenotype (Figure 2). This can be illustrated for some patients in our cohort (for instance, ID 11 and 12). At the time of presentation, they had comorbid conditions (i.e., infection, heavy menstrual bleedings) that affected clinical presentation, i.e., that worsen the anemia. They received successful treatment, and these comorbid conditions disappeared, but the anemia (although less severe) persisted.

Altogether this suggests that environmental factors such as gender, inflammatory conditions, infections, and iron restriction and requirement may contribute to the heterogeneity in IRIDA phenotype in monoallelic *TMPRSS6*-affected subjects.

Our study has several strengths. To our knowledge, this is the largest cohort of monoallelic *TMPRSS6* patients that has been described in the literature. In addition, we also included relatives as family controls in order to gain a better insight into the differences between patients and subjects. Second, for the first time, we determined the TSAT/hepcidin ratio by *standardized* measurement in all our monoallelic patients to better define whether a patient expresses the IRIDA phenotype [24]. Age- and gender-specific reference ranges of this ratio are available via our website [25].

This study also has three main limitations. First, as most patients are referred to our tertiary referral center, the majority of the patients received iron supplemental treatment at the time iron parameters were assessed and before they were diagnosed with monoallelic IRIDA. This affects the clinical presentation of IRIDA and results in diagnostic uncertainty. However, it shows real-life clinical decision-making and reveals the diagnostic delay when the treating physician is not considering IRIDA as a possible cause of iron deficiency. Moreover, an evidence-based definition of IRIDA phenotype is lacking, and in combination with the actual clinical phenotype, it can be difficult to assess the diagnosis of monoallelic IRIDA. The clinical presentation of a monoallelic IRIDA patient can be demonstrated on a sliding scale (Figure 2), in which the phenotype is the resultant of the interplay of modulating factors. This highlights the challenge for a clinician to consider IRIDA as a possible diagnosis in patients with a monoallelic genetic variant. To provide diagnostic tools, we defined IRIDA by the presence of symptoms of microcytic anemia, low TSAT, no or partial response to oral iron, and a low TSAT/hepcidin ratio (using expert opinion defined cut-off values).

Second, this is a retrospective cohort study, and we did not have data on CRP and comorbid conditions in all our patients at the time of diagnosis. However, since our patients are referred to and identified in a tertiary center, it is likely that modulating conditions (such as inflammation or other comorbidities) were excluded as a likely (main) cause in expressing the IRIDA phenotype and that they reflect *TMPRSS6* monoallelic-affected patients who present with clinically relevant anemia. Lastly, we performed haplotype analyses in probands and their relatives with the identical pathogenic exonic *TMPRSS6* variant. Although we concluded that they differ in their wild-type allele, we recommend performing haplotype analyses in a larger number of individuals and families to obtain stronger evidence of the likelihood of intra-*TMPRSS6* gene defects as the underlying cause for the difference in clinical phenotype.

## 5. Conclusions

We present a case series of 16 IRIDA patients and 8 of their relatives and show that monoallelic IRIDA is a phenotypically and genotypically heterogeneous disease entity that is more common among female patients. We explore possible causes of the underlying pathophysiology causing the IRIDA phenotype, including noncoding *TMPRSS6* variants, polygenetic inheritance, and modulating environmental factors. This leads us to suggest disease expression results from their complex interplay. Our explorative study is the first step toward a better understanding of the pathophysiological mechanisms in IRIDA patients with monoallelic exonic *TMPRSS6* variants. Further research is warranted to disentangle the contribution of different factors and enhance diagnostic accuracy.

## Figures and Tables

**Figure 1 genes-13-01309-f001:**
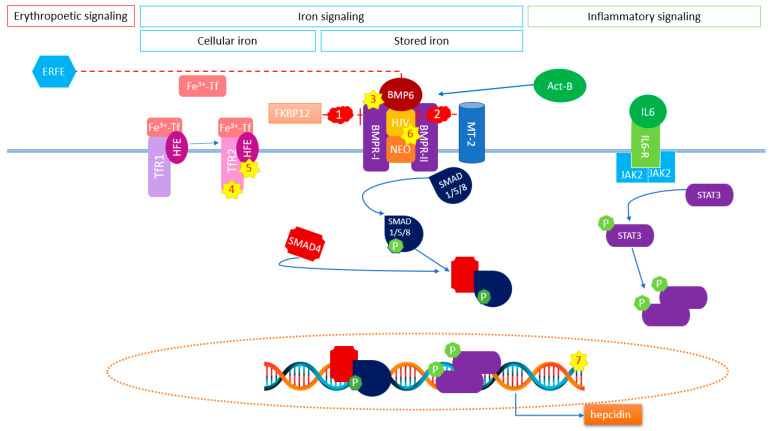
**Schematic view of hepatic hepcidin regulation by BMP/SMAD pathway and proposed potential candidate genes contributing to polygenetic inheritance in monoallelic IRIDA**. Expression of hepcidin is regulated via the BMP/SMAD pathway, which is activated by BMP receptors that bind with BMP coreceptor hemojuvelin (HJV), leading to phosphorylation and translocation of the intracellular SMAD proteins to the nucleus. Here they act as transcription factors increasing *HAMP* mRNA transcription. Hepcidin expression via this pathway is modulated by three main mechanisms: erythropoiesis, inflammation, and iron (stored and circulating). Potential candidate genes in these pathways contributing to polygenetic inheritance are indicated by numbers (1 to 7). It is conceivable that gain of function mutations of positive regulatory factors and loss of function of inhibitors of the BMP/SMAD pathway (e.g., *ACVR1* (ALK2)) can lead to an overactivation of the hepcidin response, leading to instability of the BMP pathway and, consequently, to a potential IRIDA phenotype: red clouds, loss of function (LoF); yellow stars, gain of function (GoF). Proposed genes are displayed by number: (1) *FKBP1a* (FKBP prolyl isomerase 1A) (LoF), (2) *TMPRSS6* (transmembrane serine protease 6) (LoF), (3) *BMPR1* (bone morphogenetic protein receptor type 1) (GoF), (4) *TFR2* (transferrin receptor type 2) (GoF), (5) *HFE* (homeostatic iron regulator) (GoF), (6) *HJV* (hemojuvelin) (GoF), and (7) *HAMP* (hepcidin antimicrobial peptide) (GoF). Act-B, activin B; ERFE, erythroferrone; IL-6, interleukin 6; Fe^3+^-Tf, transfferin-bound iron; TfR, transferrin receptor; JAK2, Janus kinase 2—inhibiting effect.

**Figure 2 genes-13-01309-f002:**
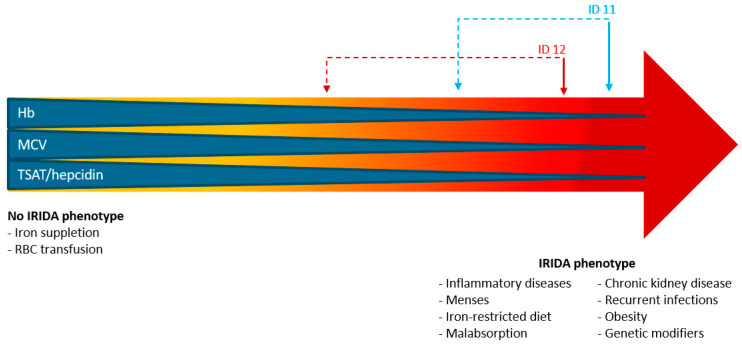
**Proposed model for the potential effect of environmental and (epi)genetic factors on IRIDA phenotype expression in *TMPRSS6* monoallelic-affected subjects**. Expression of IRIDA phenotype displayed as a sliding scale. The more to the right, the more pronounced the phenotype. Presence of modulating factors, including environmental and (epi)genetic factors, contribute to a more severe phenotype (right part of arrow). Treatment can improve phenotype (left part of arrow). Model is illustrated by two patients: ID 11, presented with Giardia lamblia infection and severe anemia. After antibiotic treatment, anemia improved but still persisted; ID 12, suffered from heavy menstrual bleedings, which stopped after starting gonadorelin antagonist. In combination with IV iron administration, IRIDA phenotype became less pronounced.

**Table 1 genes-13-01309-t001:** Characteristics of patients with a monoallelic TMPRSS6 variant and IRIDA phenotype.

ID	Sex	Age ^#^ (yrs)	Hb ^∆^ (g/dL)	MCV ^∆^ (fl)	Ferritin ^∆^ (ug/L)	TSAT ^∆^ (%)	Hb ^#^ (g/dL)	Ht ^#^ (L/L)	MCV ^#^ (fl)	MCH ^#^ (fmol)	RBC (10^12^/L)	Ferritin ^#^ (ug/L)	TSAT ^#^ (%)	TSAT/Hepcidin ^#^ (%/nM)	CRP ^#^ (mg/L)	*TMPRSS6* Variant		Iron Treatment	
																c-level	p-level	Oral	IV
																** *Deletion;Wt* **			
**1**	F	36	5.5	76	n.a.	n.a.	11.8 **	0.34	86	1.82	4.0	280	17.2	2.7	<5	c.del promotor exon1–3;Wt	p.nonsense;Wt	+	+
																** *Frameshift;Wt* **			
**2**	F	32	11.4 **	81	265	14.0	10.0	n.a.	76	n.a.	n.a.	32	4.0	0.3	no ^¥^	c.497delT;Wt	p.Leu166Argfs*37;Wt	+	+
**3 ^a^**	F	58	n.a.	n.a.	n.a.	n.a.	12.6 **	n.a.	86	n.a.	n.a.	299	13.0	0.6	n.a.	c.497delT;Wt	p.Leu166Argfs*37;Wt	+	+
**4**	F	48	12.6 *	78	136	5.0	12.1	n.a.	79	n.a.	n.a.	130	5.0	0.7	no ^¥^	c.497delT;Wt	p.Leu166Argfs*37;Wt	+	+
**5**	F	44	10.5 *	77	77	5.0	12.7 **	0.39	92	1.86	4.3	856	20.0	0.7	n.a.	c.497delT;Wt	p.Leu166Argfs*37;Wt	+	+
**6**	F	28	11.4	76	23	n.a.	11.4	0.36	71	1.38	5.1	51	5.6	0.7	n.a.	c.497delT;Wt	p.Leu166Argfs*37;Wt	+	−−
																** *Splicing;Wt* **			
**7**	M	47	12.7 *	71	n.a.	7.5	12.4 *	0.40	70	1.40	5.7	131	8.0	1.1	10	c.230-6G>A;Wt	Splicing;Wt	+	+
**8**	F	40	8.9	70	29	5.0	13.1 *	0.40	87	n.a.	4.6	198	20.0	2.6	<5	c.863+1G>T;Wt	Splicing;Wt	+	−
**9**	F	52	10.8	75	39	6.0	11.3 *	0.35	80	1.60	4.3	79	10.0	2.1	26	c.863+1G>T;Wt	Splicing;Wt	+	− ^%^
**10**	F	29	11.4	81	76	9.0	11.4 *	0.37	81	1.54	4.6	87	9.0	2.1	<1	c.431+5G>T;Wt	Splicing;Wt	+	+
																** *Missense;Wt* **			
**11 ^b^**	F	43	7.9	62	3	2.8	7.9	n.a.	62	n.a.	n.a.	30	2.7	0.5	67	c.1654G>A;Wt	p.Asp552Asn;Wt	+	+
**12 ^c^**	F	34	9.7	68	22	4.0	12.4 *	n.a.	89	n.a.	n.a.	146	12.5	2.1	1	c.2105G>T;Wt	p.Cys702Phe;Wt	+	−
**13 ^d^**	M	4	7.9	55	25	4.0	8.9 *	n.a.	55	n.a.	n.a.	16	3.0	2.2	n.a.	c.1336C>T;Wt	p.Arg446Trp;Wt	+	+
**14**	F	44	8.1	61	7	3.0	9.7	0.32	67	1.27	4.7	24	4.0	1.5	3	c.1805G>C;Wt	p.Cys602Ser;Wt	+	+
**15**	M	2	9.3	56	n.a.	n.a.	10.0	0.32	58	1.14	5.4	56	12.5	1.7	2	c.1714G>A;Wt	p.Gly572Ser;Wt	+	−
**16 ^e^**	M	3	9.7	60	50	4.0	11.1 **	0.34	74	1.52	4.6	135	8.6	0.8	1	c.1346A>G;Wt	p.Tyr449Cys;Wt	+	+
**Median**		38	10.5	71	45	5.0	11.4	0.40	78	1.50	4.6	109	8.8	1.6				100%	69%

^∆^ First available results of blood tests; ^#^ at time of genotypic diagnosis of monoallelic IRIDA; * received oral iron administration; ** received IV iron administration; ^¥^ no inflammation according to Donker et al. [3]; ^%^ intramuscular iron instead of intravenous administration. ^a^ Mother of patient 2. ^b^ Suffering from heavy blood loss due to uterine myoma at time of presentation with anemia. Despite appropriate treatment with gonadorelin antagonists to prevent menstrual bleeding and IV iron administration, IRIDA phenotype persisted. ^c^ At time of evaluation, suffering from Giardia lamblia infection. After successful antibiotic treatment, iron absorption test was still abnormal and TSAT/hepcidin remained low and, therefore, considered an IRIDA phenotype. ^d^ Suffering from Familial Mediterranean Fever (FMF), a disorder characterized by episodes of fever accompanied by serositis, synovitis, or skin rash. At time of presentation, signs of inflammation (CRP 74 mg/L). ^e^ Known with recurrent upper respiratory tract infections. At time of evaluation, no signs of infection: n.a., not available; Wt, wild-type; RBC, red blood cell count.

**Table 2 genes-13-01309-t002:** Characteristics of relatives with a monoallelic TMPRSS6 variant but without IRIDA phenotype.

ID	Sex	Relation to Proband		Age ^#^ (yrs)	Hb ^#^ (g/dL)	Ht ^#^ (L/L)	MCV ^#^ (fl)	MCH ^#^ (fmol)	RBC ^#^ (10 ^12^/L)	Ferritin ^#^ (ug/L)	TSAT ^#^ (%)	TSAT/Hepcidin ^#^ (%/nM)	CRP ^#^ (mg/L)	*TMPRSS6* Variant	
														c-level	p-level
														** *Deletion;Wt* **	
**17**	M	Son	Patient 1	8	12.7	0.37	74	1.59	5.0	53	14.1	7.1	<5	c.del promotor1-3;Wt	p.nonsense;Wt
**18**	M	Father	Patient 1	71	10.6	0.32	86	1.80	3.7	151	14.4	11.1	<5		
**19**	F	Aunt (paternal)	Patient 1	66	13.3	0.40	90	1.86	4.4	128	30.7	n.a.	<5		
														** *Missense;Wt* **	
**20**	F	Mother	Patient 12	59	14.5	0.43	95	2.00	4.5	56	21.9	12.9	1	c.2105G>T;Wt	p.Cys702Phe;Wt
**21**	F	Sister	Patient 11	38	11.9	0.35	92	1.96	3.8	68	18.3	7.3	<5	c.1654G>A;Wt	p.Asp552Asn;Wt
**22**	F	Mother	Patient 16	34	13.7	n.a.	89	n.a.	n.a.	55	28.0	n.a.	n.a.	c.1346A>G;Wt	p.Tyr449Cys;Wt
														** *Splicing;Wt* **	
**23**	F	Sister	Patient 8	54	12.9	0.37	77	1.66	4.8	88	15.0	3.1	1	c.863+1G>T;Wt	splicing;Wt
**24**	F	Mother	Patient 8	82	10.0	0.30	84	1.73	3.6	502	18.8	0.8	45		
**Median**					12.8	0.37	88	1.80	4.4	78	18.6	7.2			

^#^ At time of genotypic diagnosis of monoallelic IRIDA: n.a., not available; Wt, wild-type; RBC, red blood cell count.

## Data Availability

Data of this study are available within the article and its Appendix A.

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
