# Peer review of "IRIDA Phenotype in TMPRSS6 Monoallelic-Affected Patients: Toward a Better Understanding of the Pathophysiology"

_genes, 2022, doi:10.3390/genes13081309_

Round 1

Reviewer 1 Report

Excellent paper - very thorough review with superb discussion.

Reviewer 2 Report

Hoving et al submitted a very interesting topic discussing the possible pathogenesis of a IRIDA patients with a monoallelic IRIDAm defined as a one affected TMPRSS6 allele. The topic is very interesting to the readers and authors have a good quality of presentation. However, I have some points that might help authors to make the content more scientific and significant. 

(1)   Although you have mentioned that this study was the largest for the monoallelic TMPRSS6 patients, I believe that the number of patients (16) was very low to clearly understand the pathophysiological mechanisms in IR-389 IDA patients with monoallelic exonic TMPRSS6 variants and might not be used as a reliable reflection of the monoallelic TMPRSS6 patients with IRIDA.

(2)   You have mentioned that most IRIDA patients harbored biallelic exonic TMPRSS6, but it is better to write the approximate percentage.

(3)   Lines71-73: This sentence “Integrating phenotypic and genotypic data of our case series, we explore theoretical possible pathophysiological mechanisms underlying phenotype expression in these TMPRSS6 monoallelic affected patients” needs to be revised to be grammatically sound good. 

(4)   Table S2: I would suggest using a different color (blue or red) rather than grey for the variants that are only found in patients. 

(5)   Tables 1 and 2: as the study is retrospective, it would be better if you can include more hematological parameters, such as HCT, MCH, and RBC count for all subjects.

(6)   Line 309: “In vitro” should be italic.

Reviewer 3 Report

Dear authors,

Thanks for the opportunity to review your submission. I have a few suggestions to improve the manuscript.

-Is the word 'suppletion' on line 145 of page 3 supposed to be 'supplementation'?

-This paper would be greatly strengthened if alterations in genes other than TMPRSS6 were investigated.
